# Comparing the Physicochemical Properties of Upgraded Biomass Fuel by Torrefaction and the Ashless Technique

**Lkhagvadorj Sh [1,†], Byoung-Hwa Lee [2,†], Young-Joo Lee [3] and Chung-Hwan Jeon [1,4,*]**

1   School of Mechanical Engineering, Pusan National University, 2, Busandaehak-ro 63 beon-gil, Geumjeong-gu, Busan 46241, Korea; Lkhagva_1166@naver.com
2   Corporate R&D Institute, Doosan Heavy Industries & Construction, Volvoro 22, Seongsangu, Changwon 51711, Korea; bhlee@pusan.ac.kr
3   Korea Institute of Energy Research (KIER), 152, Gajeong-ro, Yuseong-gu, Daejeon 34129, Korea; lyj3380@kier.re.kr
4   Pusan Clean Coal Center, Pusan National University, 2, Busandaehak-ro 63 beon-gil, Geumjeong-gu, Busan 46241, Korea
*   Correspondence: chjeon@pusan.ac.kr; Tel.: +82-51-510-3051; Fax: +82-51-510-5236
†   These authors made equal contributions as first author.



**Featured Application:** Torrefaction and ashless techniques in this work are used to upgrade physicochemical properties of woody and herbaceous biomass. The ashless technique combining torrefaction process is providing promising application in herbaceous biomass specially.

**Abstract:** The purpose of this study was to investigate and compare the influence of torrefaction and an ashless process on the physical and chemical properties of pitch pine sawdust (PSD) and kenaf as types of woody and herbaceous biomass. The physicochemical properties of the materials pretreated by the ashless process with torrefaction including proximate and ultimate analysis, hydrophobicity, grindability, morphology, and structure were analyzed. The results showed that when ashless Kenaf was torrefied, the high heating rate and atomic ratios of O/C and H/C increased. The tendency of the torrefied, ashless Kenaf to absorb water decreased, and it became more hydrophobic (approximately 0% for the uptake rate of moisture). In addition, the grindability of the torrefied, ashless Kenaf was substantially improved compared to that of pretreated PSD. Brunauer–Emmett–Teller and scanning electron microscopy results showed that when Kenaf was pretreated, particles easily lost their fibrous structure and cracked as the number of macropores decreased. These results indicate that the herbaceous biomass of Kenaf, when pretreated with both torrefaction and the ashless process, exhibits improved physicochemical properties compared to the woody PSD.

**Keywords:** pitch pine sawdust; kenaf; torrefaction; ashless; grindability; hydrophobicity; chemical composition

---

## 1. Introduction

The introduction and utilization of new and renewable energy sources is increasing in modern society. Biomass will play an important role in energy supply in future energy scenarios because of the use of fossil fuels and severe global warming [1]. Of all the renewable resources, biomass is considered to be the most utilized worldwide, including in both under-developed and developed countries [2]. However, the cultivation, harvesting, drying, storage, transportation, and conversion (e.g., palletization and co-firing) of biomass require careful assessment in any fuel procurement strategy.

Recently, the development and use of large biomass sources have been hampered because biomass has been restricted for use as a solid fuel. These restrictions are primarily related to the physical and chemical properties of biomass. Compared to coal, biomass materials have a low volume density, abrasiveness, calorie content, and high moisture content. These restrictions have had a significant impact on the energy conversion efficiency of biomass materials. In addition, the entire biomass-to-energy value chain has been significantly affected by the storage, handling, and transportation of costly biomass [3–5].

Among the many pretreatment technologies available to overcome the restrictions on biomass, torrefaction is typically performed at 200–300 °C in an inert atmosphere; it is also an easy process to improve the quality of biomass feedstock. This process is usually performed at low temperatures (<200 °C) to remove water and preserve trees. However, at 200–300 °C, the weak structural component of biomass is partially decomposed; therefore, torrefaction is also referred to as mild pyrolysis. Torrefaction preserves most of the mass (>50%) and almost all the energy (approximately 90%) in raw biomass. It also improves biomass grinding by reducing energy consumption. In addition, it improves the flow characteristics and hydrophobicity of biomass, and prevents it from degrading. Thus, torrefaction is considered one of the most promising preprocessing technologies and has been widely studied as a potential component of the biomass burning and gasification process [6–9].

By torrefaction technology, the energy density and bulk density of biomass can be increased, and the cost of transportation and storage can be reduced [10]. Qing et al. indicated that torrefaction could significantly improve the physical density, energy density, and bulk density of biomass feedstock to effectively utilize storage space and reduce transportation costs. In addition, the biomass becomes brittle after torrefaction due to the breakdown of filaments in biomass by the release of gaseous and volatile products; consequently, the total pore volume and surface area of torrefied fuels is higher than that of the parent biomass [11]. The detailed mechanism of the evolution of pore structures has been explained by Onay as follows: When the temperature is relatively low (approximately 230 °C), the specific surface area and pore diameter of the torrefied fuel change little compared with raw biomass. At approximately 250 °C, the pores are enlarged and more open pores are generated due to the increased speed of volatilization of gaseous products. At the same time, volatile tar in the semi-precipitated state may plug some pores to form new pores. This effect complicates the pore structure, thereby decreasing the average pore size and increasing the specific surface area. When the temperature reached approximately 270 and 290 °C, some pores are closed and restructured, resulting in an increased average pore size and reduced specific surface area [12]. Moreover, the density and porosity according to feedstocks such as woody and herbaceous may be different because the pretreatment of biomass is heavily dependent on the degradation of the constituents in the biomass [13].

Another pretreatment method to improve biomass quality is the ash rejection process developed by Korea Institute of Energy Research (KIER). This acetic acid-based pretreatment method has been used to extract the alkali mineral inside biomass and may contribute to a significant reduction in chloride-induced corrosion and slag/fouling caused by the deposition of the boiler during combustion [14]. Lee et al. [14] proposed that the optimal pretreatment conditions for maximum ash rejection were a pH of 1.76, temperature of 60 °C, and a residence time of 10 min, at which point the losses of hemicellulose, cellulose, and lignin were minimized and investigated for several kinds of biomass.

In addition, with an aim to reduce PM emission from biomass combustion, Yani et al. [15] have proposed torrefaction and Lee et al. [16] have proposed ashless biomass pre-treatment methods. Since K, Na, and Cl are sources of PM, techniques have been developed by prior researchers to use less fuel or remove these substances.

Therefore, the ashless biomass technique used in this study is a state of the art pre-treatment method to remove ash and alkali matter in herbaceous as well as woody biomass. In addition, it is commercially applicable for the production of ashless biomass, which is seeing increasing potential as an acidic treatment of biomass compared to other pretreatment methods. To the best of our knowledge,

detailed studies on the physicochemical properties of ashless biomass produced by the above method have not been performed and compared with other pretreatment methods thus far.

Biomass can be classified into woody and herbaceous in terms of the carbon components and life cycles. Herbaceous biomass is regarded as an abundant and relatively inexpensive fuel source. Its short life cycle and low production costs imply that it can replace wood-based biomass in the market due to its high price competitiveness [17]. However, before marketing herbaceous biomass, its physical and chemical properties require further investigation. Despite recent torrefaction research that has focused on improving the fuel properties of woody biomass, very little research has thoroughly investigated and compared the structural and physicochemical properties of woody and herbaceous biomass, especially with ash rejection pretreatment. In addition, physicochemical research on herbaceous biomass pretreated by the ashless process has not been performed precisely.

The purpose of this study was to investigate not only the fuel characterization and composition of woody and herbaceous biomass pretreated by torrefaction and the ashless method, but also to determine their physical and chemical characteristics. Characterization methods included scanning electron microscopy (SEM) for morphology examination, and thermogravimetric and elemental analysis to study the changes in the O/C ratio and components in the biomass. The surface area and pore size distribution were also investigated using the Brunauer–Emmett–Teller (BET) method. Furthermore, the hydrophobicity and grindability of each type of biomass were investigated.

## 2. Materials and Methods

### 2.1. Materials

Samples of pine sawdust (PSD) and kenaf (*Hibiscus cannabinus* L.) were collected as lignocellulosic and herbaceous biomass, respectively; these are co-fired with coal in Korean power plants. The samples were placed in a grinder and sieved to sizes between 0.6 and 1.18 mm. The biomass types used in this study are raw pine sawdust (R. PSD), raw Kenaf (R. Kenaf), torrefied pine sawdust (T. PSD), torrefied Kenaf (T. Kenaf), ashless pine sawdust (A. PSD), ashless Kenaf (A. Kenaf), torrefied ashless pine sawdust (T. A. PSD), and torrefied ashless Kenaf (T. A. Kenaf), which were prepared at sizes between 0.6 and 1.18 mm using a grinder.

### 2.2. Pretreatment Methods

#### 2.2.1. Torrefaction Treatment

The torrefaction method used an inert atmosphere to thermally decompose raw biomass, and release moisture and volatile matter [6–8]. T. PSD and T. Kenaf were prepared via a torrefaction process performed under inert gas conditions. In all, 5 g of R. PSD and R. Kenaf were placed in the prepared sample crucible, following which nitrogen was introduced into the tube at 1.5 cm$^3$/min to create an inert atmosphere. The R. PSD and R. Kenaf were torrefied at 250 °C for 30 min under a nitrogen atmosphere.

#### 2.2.2. Ashless Treatment

The second procedure used was the ashless method, which involves removing the alkaline mineral matter content in ash [14]. A. PSD and A. Kenaf were produced by the ashless process with acetic acid (CH$_3$COOH). The mineral extraction was carried out in a 500 mL autoclave reactor at a heating rate of 2 °C/min until a desired temperature was reached. The samples were stirred with a magnetic stirrer at 100 rpm. After a certain reaction time, the solid residue was separated using filter paper with a pore size of 1 μm, washed continuously with ultrapure water until it reached a neutral pH, and finally oven-dried at 105 °C for 6 h. T.A. PSD and T.A. Kenaf were prepared via the torrefaction process once more at a temperature of 250 °C for 30 min under a nitrogen atmosphere after the ashless process was performed.

### 2.3. Fuel Analysis

The samples were subjected to proximate, ultimate, and calorific value analyses. Approximately 5 g of each sample for proximate analysis was analyzed by thermogravimetric analysis (TGA 701, LECO Co., St. Joseph, MI, USA) based on the ASTM D7582 testing method. The C, H, N, and S contents were measured by employing an elemental analyzer (Leco-TruSpec Micro CHNS, LECO Co., St. Joseph, MI, USA). The O content was calculated by the difference. For each sample, the proximate and ultimate analyses were repeated three to five times respectively, and average values of these measurements were presented. The gross calorific value of samples was determined using a calorimeter (AC600, LECO Co., St. Joseph, MI, USA) based on ASTM Standard D5865-03.

### 2.4. Grindability Measurements

Particle size distribution profiles for the raw and pretreated biomass were compared to that of coal for insights into their grindability. R. PSD and R. Kenaf were ground using a plate mill (Model 4E electric grinding mill, QCG Systems, Phoenixville, PA) to pass between 20 and 80 mesh with some yield down to 100 mesh. A mass of 50 g samples was ground in a mortar grinder for 15 min. The ground samples were then sieved to sizes of 0.075, 0.15, 0.3, 0.425, 0.6, 0.8 and 1.0 mm. The masses of the collected samples after each sieving were measured and recorded as a percentage of the original sample mass, and plots of the particle size distribution and cumulative particle mass of each ground sample were created.

### 2.5. Hydrophobicity Measurements

The hydrophobicity of the raw and pretreated samples were determined by immersion testing. Approximately 0.5 g of biomass samples with particle size <1 mm was immersed in deionized water at room temperature in a sintered glass filter for 2 h and air-dried for 1 h prior to determining their moisture content. The ratio of moisture uptake for samples was calculated using the differences in their weight between before and after treatment.

### 2.6. Structural Variation of Treatment Samples

A scanning electron microscope (SEM, S-3500N, Hitachi Co., Japan) was used to determine the microstructure and morphology of the samples. The sample particles were spread onto an adhesive carbon tape and placed into the SEM. The SEM was operated at the sample parameters for sample particles in the same size range. Therefore, SEM images of the sample particles are comparable in terms of size, shape, and morphology. The surface area and volume for the pores on the surface of the samples were measured using nitrogen sorption tests (all samples were degassed for 7 h at 373 K) with a Micromeritics ASAP physisorption analyzer that measures the BET surface area and pore volume in a non-destructive manneR. The surface area was calculated from the slope and intercept obtained by the BET equation. The specifications for ASAP 2020 plus physisorption (Micromeritics) are as follows: pressure measurement range: 0–950 mmHg, resolution: up to 0.1 mmHg transducer, and accuracy: ±0.1%. This system includes a 0.1 mmHg transducer and a high vacuum pump. This system can provide porosity data for micropores with a size of 0.35–3 nm as well as for pores of a larger size. Therefore, this instrument was used in the investigation of nanomaterials in this study [18–20].

## 3. Results and Discussion

### 3.1. Fuel Characterization of Treatment Samples

R. PSD and R. Kenaf contained moisture content of approximately 10%. After the ashless and torrefaction processes, all samples showed low moisture content of approximately 1% (Table 1). After torrefaction in all samples, the volatile matter (VM) decreased and the fixed carbon (FC) increased. These results are in line with findings from a previous study [21]. Torrefaction led to a decrease in VM

content and an increase in FC content. Therefore, the proximate constituents of the torrefied samples became closer to those of the coal samples. In contrast to torrefaction, A. PSD and A. Kenaf produced by the ashless process showed increase in VM and decrease in FC compared to R. PSD and R. Kenaf because of reduced ash contents and impact of ash rejection process.

**Table 1.** Fuel properties of raw and pretreated biomasses (pine sawdust (PSD) and Kenaf).

| Sample | R.P | T.P | A.P | T.A.P | R.K | T.K | A.K | T.A.K |
|---|---|---|---|---|---|---|---|---|
| Moisture (wt.%, ar) | 8.31 | 1.62 | 1.95 | 1.07 | 9.18 | 1.96 | 2.18 | 1.08 |
| Ultimate analysis (wt.%, db) | | | | | | | | |
| C | 45.79 | 51.34 | 46.24 | 52.13 | 41.70 | 49.37 | 45.56 | 51.29 |
| H | 5.78 | 5.72 | 5.91 | 5.83 | 5.47 | 4.98 | 5.96 | 5.38 |
| N | 0.08 | 0.11 | 0.06 | 0.09 | 0.63 | 0.82 | 0.56 | 0.78 |
| O [a] | 46.73 | 41.50 | 47.53 | 41.65 | 48.29 | 39.19 | 46.27 | 40.25 |
| S | 0.49 | 0.19 | 0.01 | 0 | 0.08 | 0.10 | 0.02 | 0.02 |
| Ash | 1.13 | 1.14 | 0.25 | 0.30 | 3.83 | 5.54 | 1.63 | 2.28 |
| Proximate analysis (wt.%, db) | | | | | | | | |
| VM | 79.41 | 75.26 | 82.18 | 78.87 | 76.44 | 63.36 | 80.37 | 70.85 |
| FC | 19.46 | 23.60 | 17.57 | 20.82 | 19.72 | 31.09 | 17.99 | 26.87 |
| Ash | 1.13 | 1.14 | 0.25 | 0.30 | 3.83 | 5.54 | 1.63 | 2.28 |
| HHV (kcal/kg, ar) | 4282 | 4845 | 4594 | 4807 | 4162 | 4980 | 4385 | 5053 |
| Atomic ratio (–) | | | | | | | | |
| H/C | 1.26 | 1.09 | 1.27 | 1.12 | 1.31 | 1.01 | 1.31 | 1.05 |
| O/C | 1.02 | 0.81 | 1.03 | 0.79 | 1.16 | 0.79 | 1.02 | 0.78 |

[a] Calculated by difference. ar: as received, db: dry basis, FC: fixed carbon; VM: volatile matter; HHV: higher heating value; R.P: Raw PSD, T.P: Torrefied PSD, A.P: Ashless PSD; T.A.P: Torrefied Ashless PSD; R.K: Raw Kenaf, T.K: Torrefied Kenaf, A.K: Ashless Kenaf, T.A.K: Torrefied Ashless Kenaf.

Torrefaction also caused an overall increase in the ash content, although the rates for Kenaf were higher than those for PSD. This could have been due to a loss of organic matter content in the form of VM released during torrefaction. This could result in retention of ash in the solid products, which would lead to an increase in the ash content in torrefied samples [22].

In the ashless samples, the results confirmed that more ash was removed compared to the raw sample before pre-treatment. In coal-fired power plants, ash is a major source of slagging and fouling; therefore, the ashless sample was found to be more stable than the torrefied or raw samples in terms of combustion. However, because the increase in the calorific value was small, the combined ashless-torrefied pre-treatment is determined to be the most effective process.

The results demonstrate the alterations in the elemental composition of the pretreated biomass. As expected, there was an increasing trend in carbon content after torrefaction, while oxygen and hydrogen contents decreased. The reduction in the hydroxyl groups with the formation and evolution of water vapor can explain these significant losses [23]. Further illustrations of the changes in the elemental compositions of the samples can be observed in a van Krevelen diagram (Figure 1). R. PSD and A. PSD showed similar values at higher positions, while T. PSD and T.A. PSD exhibited similar values lower than that of R. PSD. This indicates that the ashless process had a negligible impact on PSD, while the torrefaction process had significant impact on the elemental constituents.

R. Kenaf had the highest H/C and O/C values, which indicated that it contained the biomass of the lowest quality. In case of kenaf, the ashless process led to improved elemental composition, resulting in A. Kenaf being similar to existing R. PSD. However, A. Kenaf and T.A. Kenaf had lower values compared to A. PSD and T. A PSD. This suggests that when Kenaf is pretreated, carbonization can be active unlike that in pretreated PSD.

The elemental analysis of ash indicated that in the case of R.P and T.P, the carbon content increased owing to the reduction in the oxygen and moisture contents in the torrefaction process. However, in the

case of A.P, although the moisture and ash contents reduced in the manufacturing process, the increase in carbon was not large because there was no decomposition of oxygen. However, in the case of T.A.P, the highest carbon content was confirmed with the application of ash removal and torrefaction. In addition, Kenaf samples showed similar results to PSD.

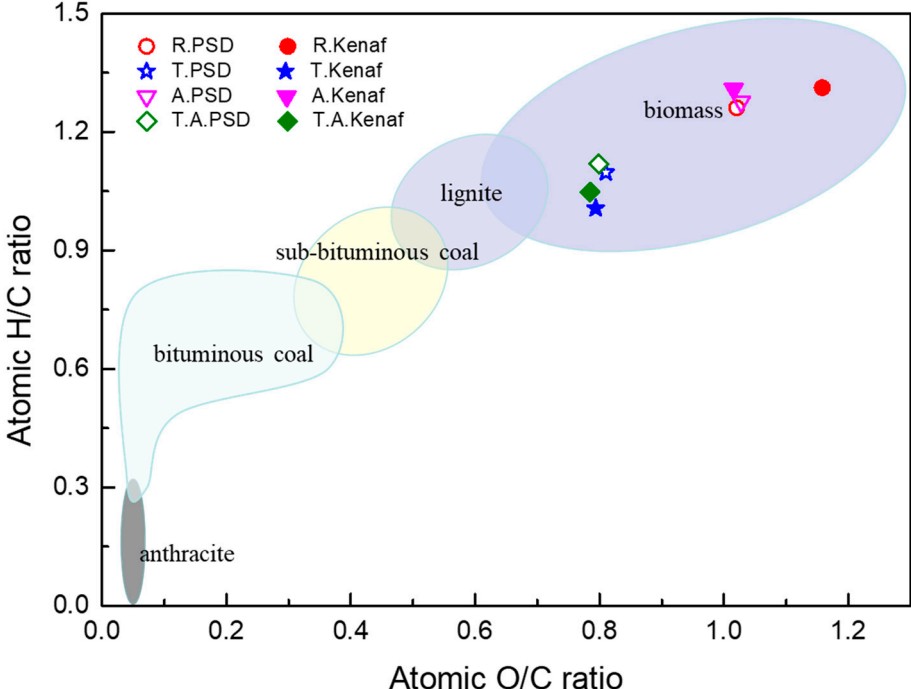

**Figure 1.** Van Krevelen diagram showing properties of raw and pretreated biomasses.

One consequence of the changing carbon, hydrogen, and oxygen contents is an increase in high heating values. After the ashless process, the higher heating value (HHV) of R. PSD and R. Kenaf increased by 7% and that of A. Kenaf increased by 5%. After torrefaction, these values for T. PSD and T. A PSD increased by 13% compared to R. PSD, while T. Kenaf and T. A. Kenaf increased by 20% compared to R. Kenaf. This indicates that the HHV of torrefied kenaf after the ashless process improved compared to other samples.

### 3.2. Structural Variation of Treatment Samples

Figure 2 shows that the pore size distributions at all regions using the BET method were based on the results of nitrogen sorption analyses of PSD samples and Kenaf samples after grinding. The porous structures are divided according to the pore diameter into micropores (those with a diameter <2 nm), mesopores (diameters 2–50 nm), and macropores (diameters >50 nm) [24]. Table 2 shows the quantitative values for pore volume and surface area for PSD samples and Kenaf samples divided by the pore diameter based on the BET results.

Following the torrefaction and ashless processes, the cumulative pore volume and surface area of T. PSD and A. PSD became larger than that of R. PSD. T. PSD showed a pore distribution similar to A. PSD. However, the A. PSD micropores are larger than those of the T. PSD. Further, the micro- and macropores of the T. A PSD became largeR. For R. Kenaf, only torrefaction was performed and a similar pore volume and surface area were observed. However, the ashless process greatly increases the distribution of the pore sizes by more than 50 nm. After torrefaction, T. Kenaf and T. A Kenaf macropores became smaller while the micro- and mesopore areas were largeR. This indicates that woody and herbaceous biomasses were influenced differently by the ashless and torrefaction processes.

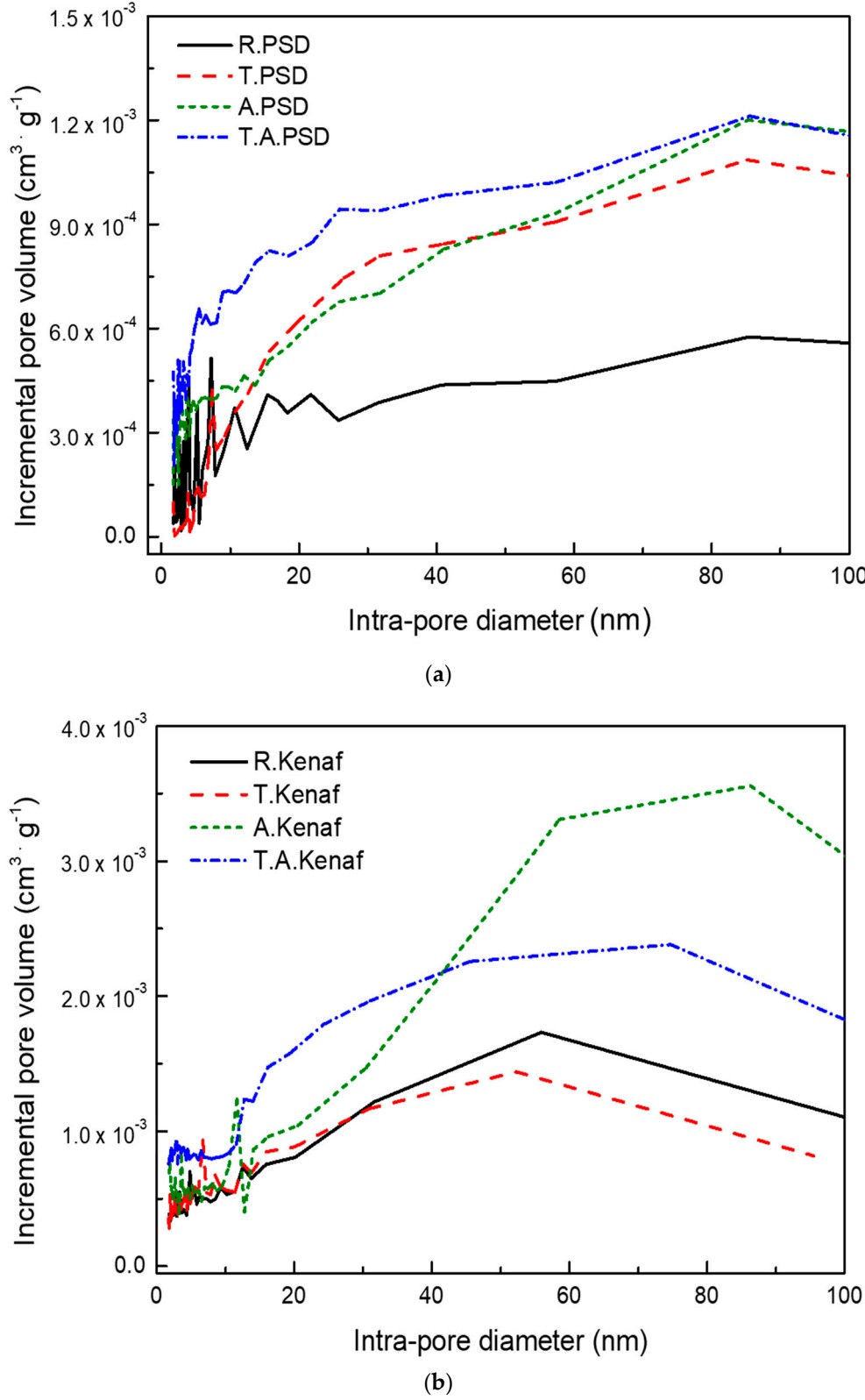

(**a**)

(**b**)

**Figure 2.** Pore size distribution of PSD samples (**a**) and kenaf samples (**b**) at all regions by nitrogen sorption analysis.

**Table 2.** Quantitative values for pore volume and surface area of PSD samples (**a**) and kenaf samples (**b**) by nitrogen sorption analysis.

| (a) | | | | |
|---|---|---|---|---|
| | **R.PSD** | **T.PSD** | **A.PSD** | **T.A.PSD** |
| Cumulative pore volume (cm$^3$/g) | $8.3 \times 10^{-4}$ | $1.32 \times 10^{-3}$ | $1.58 \times 10^{-3}$ | $1.88 \times 10^{-3}$ |
| Micro (<2 nm) | $1.25 \times 10^{-5}$ | $3.60 \times 10^{-6}$ | $1.95 \times 10^{-5}$ | $2.93 \times 10^{-5}$ |
| Meso (2–50 nm) | $3.79 \times 10^{-4}$ | $5.43 \times 10^{-4}$ | $6.79 \times 10^{-4}$ | $9.69 \times 10^{-4}$ |
| Macro (>50 nm) | $4.39 \times 10^{-4}$ | $7.74 \times 10^{-4}$ | $8.86 \times 10^{-4}$ | $8.85 \times 10^{-4}$ |
| Cumulative surface area (m$^2$/g) | 0.19 | 0.174 | 0.39 | 0.541 |
| Micro (<2 nm) | 0.026 | 0.008 | 0.042 | 0.063 |
| Meso (2–50 nm) | 0.147 | 0.136 | 0.314 | 0.444 |
| Macro (>50 nm) | 0.016 | 0.030 | 0.033 | 0.034 |
| (b) | | | | |
| | **R.Kenaf** | **T. Kenaf** | **A. Kenaf** | **T.A. Kenaf** |
| Cumulative pore volume (cm$^3$/g) | $1.97 \times 10^{-3}$ | $1.86 \times 10^{-3}$ | $3.16 \times 10^{-3}$ | $3.10 \times 10^{-3}$ |
| Micro (<2 nm) | $1.73 \times 10^{-5}$ | $2.83 \times 10^{-5}$ | $4.88 \times 10^{-5}$ | $7.33 \times 10^{-5}$ |
| Meso (2–50 nm) | $9.36 \times 10^{-4}$ | $9.51 \times 10^{-4}$ | $1.30 \times 10^{-3}$ | $1.97 \times 10^{-3}$ |
| Macro (>50 nm) | $1.02 \times 10^{-3}$ | $8.79 \times 10^{-4}$ | $1.81 \times 10^{-3}$ | $1.06 \times 10^{-3}$ |
| Cumulative surface area (m$^2$/g) | 0.528 | 0.568 | 0.731 | 0.997 |
| Micro (<2 nm) | 0.037 | 0.062 | 0.105 | 0.159 |
| Meso (2–50 nm) | 0.433 | 0.456 | 0.541 | 0.793 |
| Macro (>50 nm) | 0.055 | 0.05 | 0.085 | 0.045 |

The herbaceous biomass had about twice as much the pore distribution as the woody biomass and the ashless process generated more pores in all regions of the samples, with an increase in macropores in particulaR. Torrefaction, in the case of PSD, showed an increase in pore sizes at all regions, including macropores. In contrast, the macropores in Kenaf were reduced while the micro- and mesopore regions were enlarged.

The different pore size distributions of woody and herbaceous biomasses can be explained by the results of Chen et al. [10]. They indicated that the degradation of agricultural residues is greater than that of ligneous plants because of the higher volatile and hemicellulose contents in herbaceous residues; they also showed that the weight loss of herbaceous biomass with a higher xylan content is more likely to occur compared to woody biomass owing to the different chemical structure. Xylan is more reactive and can break down more easily at a low temperature compared to glucomannan [25]. In addition, the rapid release of volatile matter can open and link the blind and closed pores during torrefaction and create new cracks, micropores, and mesopores, which can lead to a significant increase in porosity in case of woody biomass. In contrast, a greater amount of degradation of hemicellulos in the case of herbaceous biomass leads to a decrease in macropores.

SEM images of particles were taken and measured to observe these differences between PSD and kenaf more clearly.

SEM images of PSD and kenaf showed that the torrefaction and ashless processes had differing degrees of influence (Figure 3). Within R. PSD, very strong, bulky xylem tissues were observed. Following torrefaction, PSD began to lose it bound, fibrous structure, and cracks and large pores became more obvious in the particles. The structure of A. PSD by the ashless process appeared to be more broken, and the T. A PSD by torrefaction further facilitated this phenomenon.

The kenaf images contained a different microstructure and morphology compared to PSD. Particles from torrefied kenaf had smoother and cleaner surfaces compared to those from R. Kenaf. Additionally, T. A Kenaf contained a greater degree of decomposition and cracking in the particles. Thus, kenaf particles appeared to lose their characteristic fibrous structure and became smoother compared to PSD after the torrefaction and ashless processes, as confirmed by the BET results.

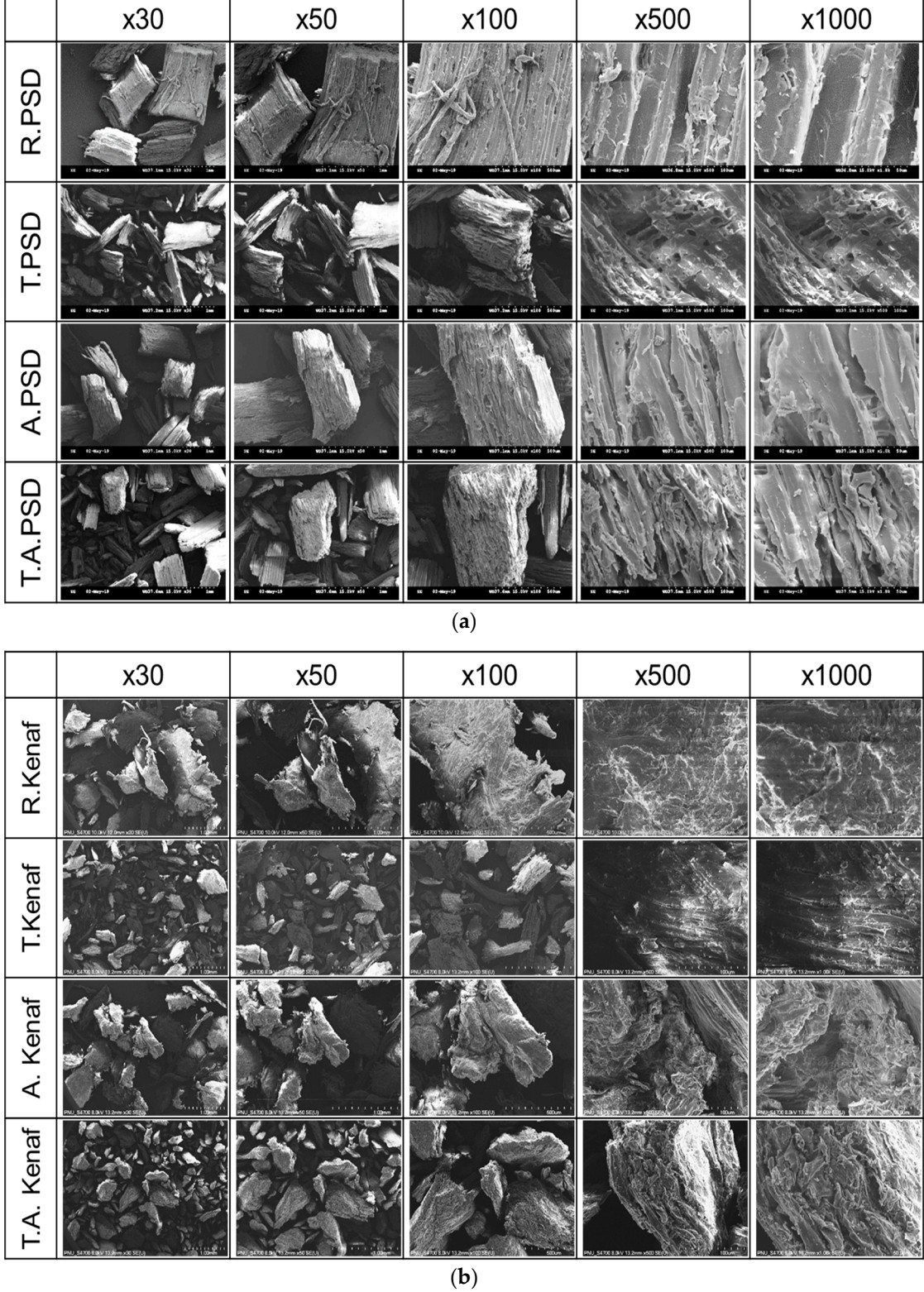

**Figure 3.** SEM images of PSD samples (**a**) and kenaf samples (**b**).

### 3.3. Comparison of Hydrophobicity for Treatment Samples

Kenaf has higher hygroscopicity when compared with PSD and there is a general trend for the reduction of absorbed moisture content after torrefaction for both PSD and Kenaf (Figure 4). Both R. PSD

and R. Kenaf experienced the highest uptake of water during the two hours of immersion (41% and 81%, respectively) and uptake of water for R. Kenaf was higher than that of R. PSD, while A. PSD and A. Kenaf absorbed considerably lesser water than that of raw samples (36% and 67%, respectively). After torrefaction of raw samples, the uptake of water was much lower (2% and 9%, respectively) and torrefied; further, ashless samples absorbed the least amount of water (approximately 0%). These findings proved that torrefaction improved the hydrophobicity of both woody and herbaceous biomass. This type of torrefaction impact was in agreement with results of Chen et al. [26]. Water is absorbed by the woody materials owing to the presence of hydroxyl groups (–OH) that attract and hold water molecules through hydrogen bonding. In wood materials, hemicelluloses are more hydrophilic than cellulose and lignin. Hemicelluloses and the noncrystalline region of cellulose chains can attract water easily, owing to the availability of hydroxyl groups [27]. The carboxylic acid groups (–COOH) in hemicelluloses are also active in absorbing water [28]. However, the degradation of hemicelluloses and amorphous cellulose by torrefaction results in the removal of –OH and –COOH groups and further decreases hydrogen bonding with wateR. As a result, tar condenses inside the pores, thereby obstructing the passage of moist air through the solid, avoiding water vapor condensation. The polar character of condensed tar on the solid also prevents the condensation of water vapor inside the pores [29].

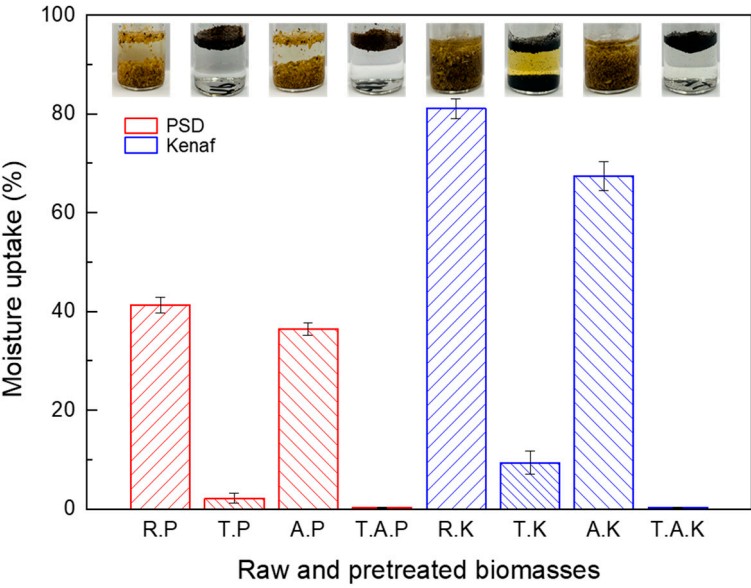

**Figure 4.** Hydrophobic properties of raw and pretreated biomasses.

Here, evidently, the uptake rate of water (9%) and reduction rate of water (88%) after torrefaction for kenaf is higher than that of PSD, which means that hydrophobicity of kenaf after torrefaction improved. Most importantly, it was observed that hydrophobicity of torrefied ashless samples was markedly improved in both T. A PSD and T. A Kenaf, which indicated that ashless process with torrefaction lead to further hydrophobic characteristics. These improvements can efficiently enhance the storage and transport of biomass samples, particularly different types of herbaceous. Alternatively, the prolonged storage life of biomass implies the enhancement of carbon sequestration ability.

## 3.4. Comparison of Grindability for Treatment Samples

The particle size distribution of PSD and Kenaf showed a high distribution in region of 0.6–0.425 mm and 0.85–0.6 mm (Figure 5). Among them, the largest particle size distribution of 1.0–1.18 mm or more had a large distribution of R. PSD and R. Kenaf, as well as a smaller distribution of T. PSD, A. PSD, and T. A PSD. As the torrefaction process, the smaller particle distribution is higheR. This is

indicated that torrefaction lead to improved grindability as shown in the torrefaction of Kenaf samples. This trend is consistent with those reported in similar studies investigating the grinding of torrefied biomass [30]. This indicates that during torrefaction, continuous decomposition of hemicellulose causes weakening and destruction of the highly interlinked cellulose–hemicellulose matrix, which no longer can support the cellulose fibers [31]. In addition, T. Kenaf, A. Kenaf, and T. A Kenaf are mainly distributed in areas below the smallest size distribution (0.15–0.075 mm). Thus, kenaf shows a smaller size distribution than that of PSD when samples are pretreated.

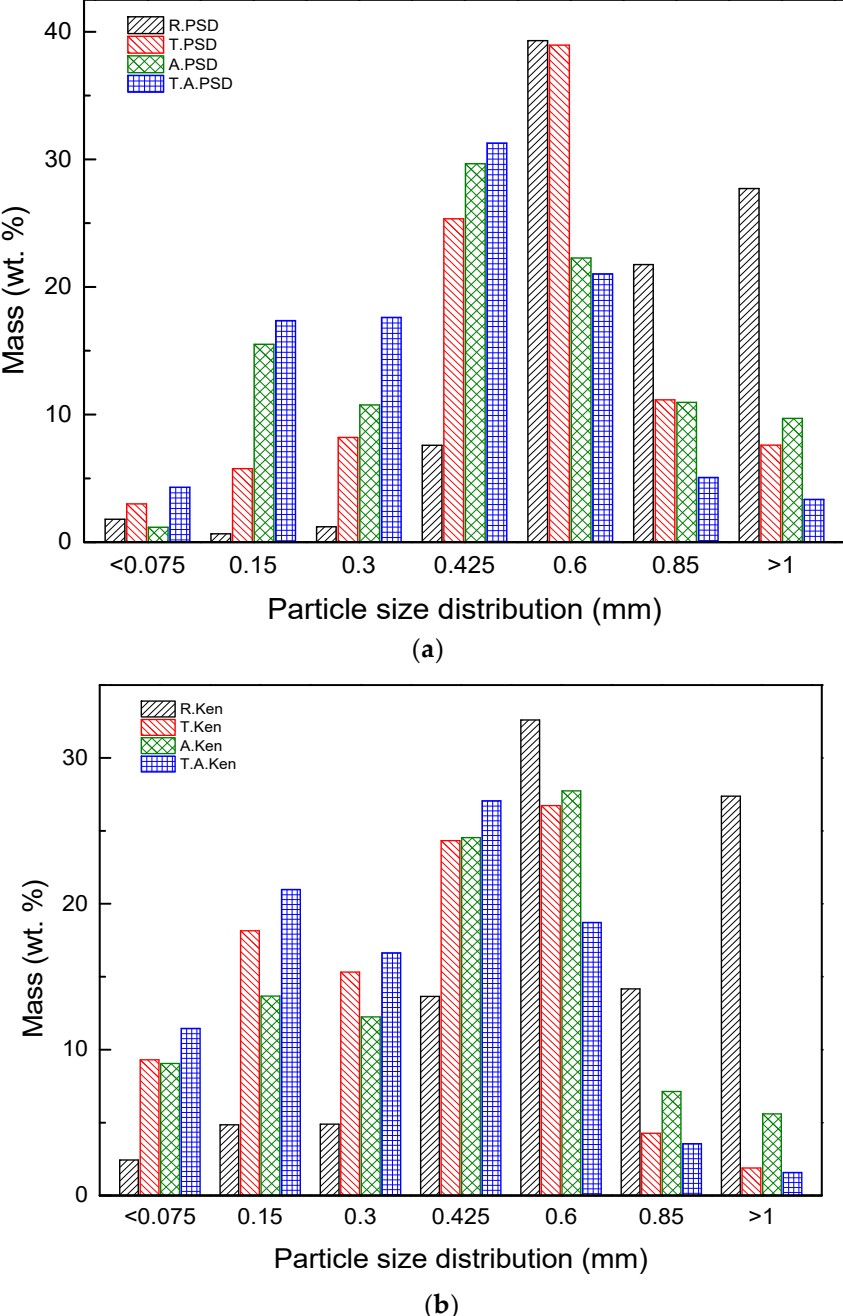

**Figure 5.** Particle size distributions of PSD samples (**a**) and kenaf samples (**b**)'s raw and pretreated biomasses.

When examining cumulative particle mass (%), the slowest rising curve was observed in R. PSD, followed by R. Kenaf, which implies that R. Kenaf had better grindability than that of raw material

(Figure 6). For pretreated PSD, torrefaction led to improved grindability and the ashless process showed further improvement. The curves of cumulative particle mass clearly shifted towards smaller particles. Similar changes of particle size distribution of ground biomass have been reported in other studies [7].

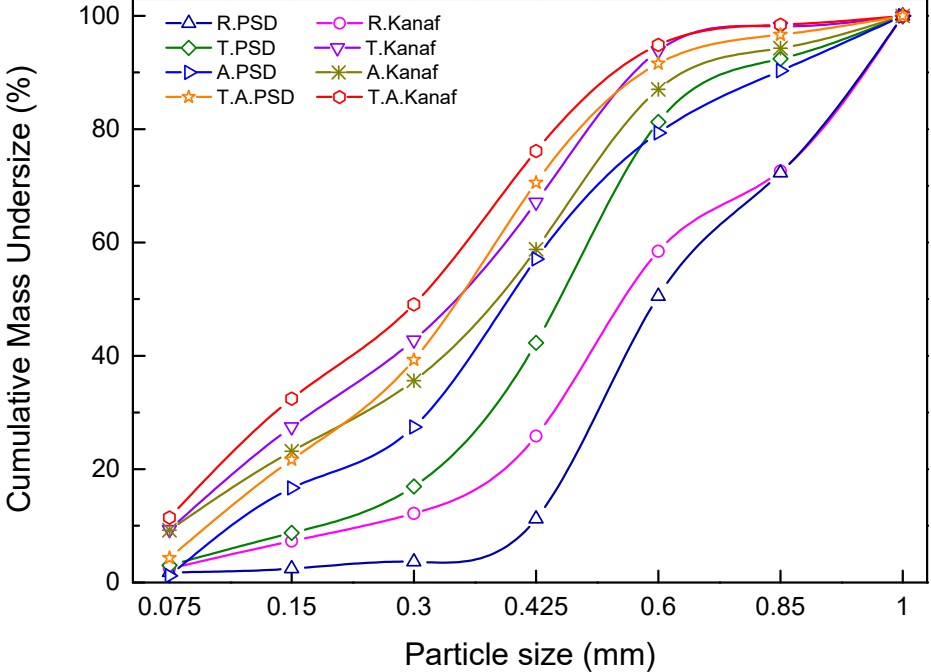

**Figure 6.** Cumulative particle mass of raw and pretreated biomasses.

T. A PSD shows better grindability than that of A. PSD. The same trend is observed in pretreated kenaf. However, the pretreatment of kenaf resulted in a large increase in grindability. To further examine this point, the cumulative particle mass by the range of 0.075 ~ 0.6 mm was compared with each other (Figure 7). R. PSD comprised of 11% of the total particles, while R. Kenaf comprised of 26%. This indicates that the smaller particle sizes of R. Kenaf were distributed more in this range. For pretreated PSD, the values were raised in the order T. PSD, A. PSD, and T. A PSD, with respective values of 42%, 57%, and 70%. The same trend was observed in the pretreated kenaf; however, the values increased rapidly to 67%, 58%, and 76%, respectively, and the rate of grindability also increased. This indicates that pretreated kenaf had a greater grindability characteristic than that of PSD.

Torrefaction and ashless processes of biomass led to the decomposition and breakdown of the hemicellulose–cellulose interlinked matrix caused by heating and the chemical treatment. This considerably simplifies the grinding of pre-treatment samples. For torrefaction biomass, the degradation of the hemicellulose and cellulose components is larger through the pre-treatment process. As a result, torrefaction biomass has a greater grindability compared with ashless biomass.

Featured Application: torrefaction and ashless techniques in this work were used to upgrade physicochemical properties of woody and herbaceous biomass. The ashless technique combining torrefaction process provided a promising application in herbaceous biomass especially.

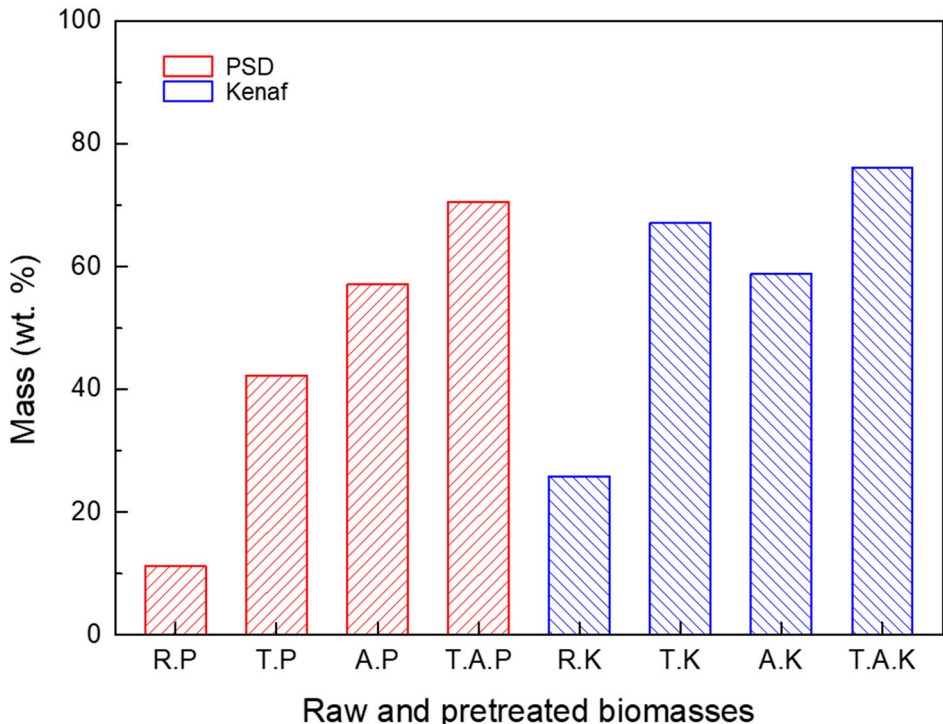

**Figure 7.** Cumulative particle mass by range of 0.6 mm of raw and pretreated biomasses.

## 4. Conclusions

The physical and chemical characteristics of lignocellulosic (PSD) and herbaceous biomass (kenaf) pretreated with torrefaction and ashless process were investigated and compared. The conclusions were as follows:

The torrefaction and ashless processes of PSD and kenaf improved the fuel ratio, with decreased volatile matter and increased fixed carbon and heating values, compared with that of untreated biomasses; further, the atomic ratios of O/C and H/C decreased. When kenaf was pretreated with both torrefaction and ashless processes, the increasing rate of HHV and atomic ratios of O/C and H/C improved compared to pretreated PSD, suggesting that herbaceous biomass was favorable to be carbonized.

The results of BET show that the torrefaction and ashless processes for R. PSD and R. Kenaf generally led to a more porous structure. However, for R. Kenaf, macropores were reduced after pretreatment, implying that kenaf particles lost their fibrous structure more easily and became more cracked compared with R. PSD. These results were also confirmed by SEM images.

R. Kenaf was more hydrophilic than R. PSD. However, when pretreatment was conducted, hydrophilicity improved and became hydrophobic. Both samples were excellent when the torrefaction and ashless processes were conducted simultaneously, indicating that combining the ashless process with torrefaction lead to further hydrophobic characteristics.

The grindability of the samples was assessed by measuring the particle size distribution after grinding. The results demonstrated the R. Kenaf was easier to grind compared with R. PSD. Combining the torrefaction and ashless processes lead to increased grindability. When kenaf was pre-treated, the effect for grindability was much greater.

In conclusion, the torrefaction and ashless processes could considerably improve the physicochemical properties in terms of HHV, hydrophobicity, and grindability. In particular, when the ashless fuel was torrefied again, the effect was confirmed to be very large. When comparing PSD and

kenaf, the physical and chemical characteristics of kenaf for herbaceous were better than those of PSD for woody.

**Author Contributions:** L.S., B.-H.L. and C.-H.J. conceived and planned the research. L.S., carried out all experiments for this research. L.S., Y.-J.L. and B.-H.L. contributed to the analysis of the results. L.S., B.-H.L. and C.-H.J. contributed to the review of original and revised papeR. L.S. and B.-H.L. took the lead in writing the manuscript. All authors helped shape the research and discussed the results and contributed to the final manuscript.

**Funding:** This research received no external funding.

**Acknowledgments:** This work was supported by the Human Resources Development program (No. 20184030202060) of the Korea Institute of Energy Technology Evaluation and Planning (KETEP) grant funded by the Korea government Ministry of Trade, Industry and Energy.

**Conflicts of Interest:** The authors declare no conflict of interest.

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
