# Peer review of "Comparing the Physicochemical Properties of Upgraded Biomass Fuel by Torrefaction and the Ashless Technique"

_applsci, doi:10.3390/app9245519_

Round 1

Reviewer 1 Report

See file in attachment

Author Response

Review on the article

“Comparing the Physicochemical Properties of Upgraded Biomass Fuel by Torrefaction and the Ashless Technique”

General comment _1

This article presents a comparative study of two pretreatment of two biomasses: the torrefaction and the ashless technique developed by the Korea Institute of Energy Research which aim to reduce mineral content of the biomass and limit corrosion during combustion.

Authors have conducts different analyses to study the modifications induced by these treatments on the raw biomasses. Experiments chosen are relevant for the study. However many precisions are missing in the materials and methods section, and imprecisions and errors are presents in the results part. This needs to be corrected before publication in applied sciences journal.

Specifics comments

Introduction:

In your article you are opposing the lignocellulosic biomasses (from wood) and herbaceous biomass. If this opposition is currently done in ordinary usage, it is a mistake. The is a confusion between ligneous material as wood, and lignocellulosic biomass which is a biomass composed of lignin, cellulose and hemicellulose. In this case, the two biomasses are lignocellulosic biomasses. In Kenjac, there is approximatively 18 % of lignine. (See for example ref: Ken-ichi Kuroda, Akiko Izumi, Bibhuti B Mazumder, Yo7582shito Ohtani, Kazuhiko Sameshima, , Journal of Analytical and Applied Pyrolysis, Volume 64, Issue 2, 2002, Pages 453-463, ISSN 0165-2370, https://doi.org/10.1016/S0165-2370(02)00047-5.) è We appreciate your valuable comments for our manuscript. We agree your opinion for usage of lignocellulosic word. As per your comment and following reference, we have modified the lignocellulosic biomass to woody biomass on Pages 1, 2, 3, 7, 11, 15.[References]  è We appreciate your valuable comments for our manuscript. As per your comment, we have added influence on density of the materials related to closed porosity on Pages.è “Using By torrefaction technology, the energy density and bulk density of biomass can be increased, and the cost of transportation and storage can be reduced [10]. Qing et al. indicated that torrefaction could significantly improve the physical density, energy density, and bulk density of biomass feedstock to effectively utilize storage space and reduce transportation costs. In addition, the biomass becomes brittle after torrefaction due to the breakdown of filaments in biomass by the release of gaseous and volatile products; consequently, the total pore volume and surface area of torrefied fuels is higher than that of the parent biomass [11]. The detailed mechanism of the evolution of pore structures has been explained by Onay as follows: When the temperature is relatively low (approximately 230 ℃), the specific surface area and pore diameter of the torrefied fuel change little compared with raw biomass. At approximately 250 ℃, the pores are enlarged and more open pores are generated due to the increased speed of volatilization of gaseous products. At the same time, volatile tar in the semi-precipitated state may plug some pores to form new pores. This effect complicates the pore structure, thereby decreasing the average pore size and increasing the specific surface area. When the temperature reached approximately 270 and 290 ℃, some pores are closed and restructured, resulting in an increased average pore size and reduced specific surface area [12]. Moreover, the density and porosity according to feedstocks such as woody and herbaceous may be different because the pretreatment of biomass is heavily dependent on the degradation of the constituents in the biomass [13].”[References to be added] [11] Yanqing Niu, Yuan Lv, Yu Lei, Siqi Liu, Yang Liang, Denghui Wang, Shi'en Hui, Biomass torrefaction: properties, applications, challenges, and economy, Renewable and Sustainable Energy Reviews 115 (2019) 109395[13] Jian Deng, Gui-jun Wang, Jiang-hong Kuang, Yun-liang Zhang, Yong-hao Luo, Pretreatment of agricultural residues for co-gasification via torrefaction, Journal of Analytical and Applied Pyrolysis, 2009, 86(2), 331-337 [12] Ozlem Onay, Influence of pyrolysis temperature and heating rate on the production of bio-oil and char from safflower seed by pyrolysis, using a well-swept fixed-bed reactor, Fuel Processing Technology 2007, 88(5), 523-531 [10] CHEN Qing, ZHOU JinSong, LIU BingJun, MEI QinFeng, LUO ZhongYang, Influence of torrefaction pretreatment on biomass gasification technology, Chinese Science Bulletin, May 2011, Volume 56, Issue 14, pp 1449–1456     One of the main differences between wood product and herbaceous biomass that can influence your study could be the density of the materials related to closed porosity. This point is not discussed in your article. You should give this information in your introduction from literacy data or in the result part based on measurement. Yanqing Niu, Yuan Lv, Yu Lei, Siqi Liu, Yang Liang, Denghui Wang, Shi'en Hui, Biomass torrefaction: properties, applications, challenges, and economy, Renewable and Sustainable Energy Reviews 115 (2019) 109395    

You said in your introduction that the used of biomass requires a careful assessment in anyfuel procurement strategy (line 37) and you compare in your study 2 pretreatements. In particular, the ashless technique is not without impact on the environment (use of solvent at pH 1.76 and intensive drying step after the pretreatement). You should provide in your introduction information about the environmental impact of this two pretreatments. Do you have an idea of the energetic cost of each of these pretreatements?

è We appreciate your valuable comments for our manuscript. We agree with the reviewer’s opinion. The introduction added PM impact for pre-treatment methods such as torrefaction and ashless biomass.

è In addition, with an aim to reduce PM emission from biomass combustion, Yani et al. [15] have proposed torrefaction and Lee et al. [16] have proposed ashless biomass pre-treatment methods. Compared with biomass before pre-treatment, torrefaction pre-treatment can reduce alkali minerals such as Na, K, and Cl under the condition of similar calorific values, whereas the ashless biomass technique can directly remove alkali minerals at the original.

[References to be added]

[15] Syamsuddin Yani, Xiangpeng Gao, and Hongwei Wu, Emission of Inorganic PM10 from the Combustion of Torrefied Biomass under Pulverized-Fuel Conditions, Energy Fuels 2015, 29, 800−807

[16] Young-Joo Lee, Ju-Hyoung Park, Gyu-Seob Song, Hueon Namkung, Se-Joon Park, Joeng-Geun Kim, Young-Chan Choi, Chung-Hwan Jeon, Jong Won Choi, Characterization of PM2.5 and gaseous emissions during combustion of ultra-clean biomass via dual-stage treatment, Atmospheric Environment 193 (2018) 168–176

Materials and methods:

General organization of the section: In the material and methods sections, you must dissociated “fuel characterization” and “proximate and ultimate analyses”. You are using an ashless technique as a pretreatment so you must compare the quantity of ash in all samples. This information is missing in your work. It will provide information about the efficiency of this pretreatment for both biomasses.è “In the ashless samples, the results confirmed that more ash was removed compared to the raw sample before pre-treatment. In coal-fired power plants, ash is a major source of slagging and fouling; therefore, the ashless sample was found to be more stable than the torrefied or raw samples in terms of combustion. However, because the increase in the calorific value was small, the combined ashless-torrefied pre-treatment is determined to be the most effective process.”   è We agree with the reviewer’s comment. In results and discussion chapter, we added the ash characterization of ashless technique. Sections “2. 1 Materials and 2.2.1 Torrefactions”: Paragraphs need to be reformulated. You write twice the same thing in line 84 and line 88. Sentence was not corrected and. Same thing in line 94 and 96è “ In all, 5 g of R. PSD and R. Kenaf were placed in the prepared sample crucible, following which nitrogen was introduced into the tube at 1.5 cm3/min to create an inert atmosphere.”   è We agree with the reviewers that the content of "Sections 2.1 Materials and 2.2.1 Torrefactions" is similar. The previous sentence under lines 94 and 96 is corrected. Section Grindability. Specify the type of mill.è “R. PSD and R. Kenaf were ground using a plate mill (Model 4E electric grinding mill, QCG Systems, Phoenixville, PA) to pass between 20 and 80 mesh with some yield down to 100 mesh.”   è We appreciate your valuable comments about our statements. As per your comments, we changed specify the type of mill. Furthermore, additional explanation is given to revised manuscript as follows, Section Morphology and chemical composition:

- The title of the paragraph is not relevant for the section. The SEM and BET analyses are not chemical analyses.

è We appreciate your valuable comments about our statements. As per your comments, we revised title as follows,

è Morphology and chemical composition è Structure variation of treatment samples

- Specify the type of ASAP used for measurement. Indeed the porous distribution, in particular the information concerning micropores requires the use of specific sensors which are not available on all Micromeritics devices. Explain how you obtain the porous distribution from the analysis, or give a reference.

è We appreciate your valuable comments for our manuscript. The specifications for ASAP 2020 plus physisorption (Micromeritics) are as follows; Pressure measurement: 0 to 950 mmHg, Resolution: up to 0.1 mmHg transducer, Accuracy: ± 0.1%. This system includes a 0.1 mmHg transducer and a high vacuum pump. It is confirmed that this system is providing porosity data on micropores between 0.35 and 3 nanometers as well as larger pore sizes. Therefore, this instrument has been used in the investigation of nanomaterials in this study. The pores measured by this instruments are classified according to diameter where micropores have diameters less than about 2 nm, mesopore sizes range from about 2 nm to about 50 nm and macropores have diameters greater than about 50 nm.

è Following sentences have added on page 4 in revised manuscript. “The specifications for ASAP 2020 plus physisorption (Micromeritics) are as follows: pressure measurement range: 0 to 950 mmHg, resolution: up to 0.1 mmHg transducer, and accuracy: ± 0.1%. This system includes a 0.1 mmHg transducer and a high vacuum pump. This system can provide porosity data for micropores with a size of 0.35 to 3 nm as well as for pores of a larger size. Therefore, this instrument was used in the investigation of nanomaterials in this study [18-20].”

è Following brochure for ASAP 2020 plus physisorption and literatures published by using this instruments as follow;

https://www.micromeritics.com/product-showcase/ASAP-2020-Plus-Physisorption.aspx

[References to be added]

[18] Arash Arami-Niya, Wan Mohd Ashri Wan Daud, Farouq S. Mjalli, Using granular activated carbon prepared from oil palm shell by ZnCl2 and physical activation for methane adsorption, Journal of Analytical and Applied Pyrolysis, 2010, 89(2) 197-203

[19] Buyi Li, Xin Huang, Liyun Liang and Bien Tan, Synthesis of uniform microporous polymer nanoparticles and their applications for hydrogen storage, J. Mater. Chem., 2010, 20, 7444–7450

[20] Arash Arami-Niy, Wan Mohd Ashri Wan Daud, Farouq S. Mjalli, Comparative study of the textural characteristics of oil palm shell activated carbon produced by chemical and physical activation for methane adsorption, Chemical Engineering Research and Design, 2011, 89(6) 657-664

Section results

Line 144, explain FC (Fixed carbon) before introducing the abbreviationFC è fixed carbon (FC) è As per your comments, we have modified the font of “FC” in line 144 As explained previously, the ultimate analysis must also be discussed with data concerning ash content in both biomasses. This information must be added.  è We appreciate your valuable comments about our statements. As per your comments, we have added comparing the correlation between elemental analysis and ash.

Sample

R.P

T.P

A.P

T.A.P

R.K

T.K

A.K

T.A.K

Moisture (wt.%, ar)

8.31

1.62

1.95

1.07

9.18

1.96

2.18

1.08

Ultimate analysis (wt.%, db)

C

45.79

51.34

46.24

52.13

41.70

49.37

45.56

51.29

H

5.78

5.72

5.91

5.83

5.47

4.98

5.96

5.38

N

0.08

0.11

0.06

0.09

0.63

0.82

0.56

0.78

Oa

46.73

41.50

47.53

41.65

48.29

39.19

46.27

40.25

S

0.49

0.19

0.01

0

0.08

0.10

0.02

0.02

Ash

1.13

1.14

0.25

0.30

3.83

5.54

1.63

2.28

“The elemental analysis of ash indicated that in the case of R.P and T.P, the carbon content increased owing to the reduction in the oxygen and moisture contents in the torrefaction process. However, in the case of A.P, although the moisture and ash contents reduced in the manufacturing process, the increase in carbon was not large because there was no decomposition of oxygen. However, in the case of T.A.P, the highest carbon content was confirmed with the application of ash removal and torrefaction. In addition, Kenaf samples showed similar results to PSD.”

Line 180: you write “the ore size distribution instead of “pore size distribution”è the ore size distribution è the pore size distribution   è As per your comments, we have modified the font of “the ore size distribution” in line 180 Morphology and BET results. The impact of pretreatment on the porosity of the samples must be discussed in relation to the density of the materials and the closed porosity. It’s probable that the pretreatment “open” a closed porosity.  [10] CHEN Qing, ZHOU JinSong, LIU BingJun, MEI QinFeng, LUO ZhongYang, Influence of torrefaction pretreatment on biomass gasification technology, Chinese Science Bulletin, May 2011, Volume 56, Issue 14, pp 1449–1456  [25] Mark J. Prins, Krzysztof J. Ptasinski, Frans J.J.G. Janssen, More efficient biomass gasification via torrefaction, Energy 31 (2006) 3458–3470 [References to be added] è Following sentences have added on page 7 in revised manuscript. “The different pore size distributions of woody and herbaceous biomasses can be explained by the results of Chen et al [10]. They indicated that the degradation of agricultural residues is greater than that of ligneous plants because of the higher volatile and hemicellulose contents in herbaceous residues; they also showed that the weight loss of herbaceous biomass with a higher xylan content is more likely to occur compared to woody biomass owing to the different chemical structure. Xylan is more reactive and can break down more easily at a low temperature compared to glucomanna. [25] In addition, the rapid release of volatile matter can open and link the blind and closed pores during torrefaction and create new cracks, micropores, and mesopores, which can lead to a significant increase in porosity in case of woody biomass. In contrast, a greater amount of degradation of hemicellulos in the case of herbaceous biomass leads to a decrease in macropores.” è We appreciate your valuable comments for our manuscript. As per your comments, the Morphology and BET results was influenced by feedstock types such as woody and herbaceous biomass. Following sentences have added to discuss in relation to the density of materials and the closed porosity according to feedstock types. Line 234 : What do you mean by “to improve the physical property of both lignocellulosic biomasses”è “These findings prove that torrefacition improves the hydrophobicity of both woody and herbaceous biomass.”   è We appreciate your valuable comments for our manuscript. As per your comments, sentences have modified as follows; Line 257 : “the fig 5 formed a particle size distribution and not the Rosin-Rammer”. The Rosin- Rammer is a model that can fit a particle size distribution. If you say that the fig 5 forms a Rosin-Rammer model, you must demonstrate it and give the fitting parameters  è We appreciate your valuable comments for our manuscript. As per your comments, that expression for rosin-rammer have removed in order to avoid misunderstanding. Line 269 what do you mean by which implies that R. Kenaf is better for raw material grindability”?è Following sentences have added on page 12 in revised manuscript. “R. Kenaf has better grindability than that of raw material.”   è We appreciate your valuable comments for our manuscript. As per your comments, the sentence have revised as follows; Both pretreatments induce chemical changes in the raw materials but they also induce physical modifications. You cannot directly relate the grindability of a materials to its chemical composition. A lot of raw materials have similar composition but very different grindability due to their physical properties and their histological structure which generally induce anisotropy. For example; graphite and diamont are both composed of carbon but do not have the same grindability. Your discussion must integrate the physical modifications induced by pretreatment  è Following sentences have added on page 12 in revised manuscript. “Torrefaction and ashless processes of biomass lead to the decomposition and breakdown of the hemicellulose-cellulose interlinked matrix caused by heating and chemical treatment. This considerably simplifies the grinding of pre-treatment samples. For torrefaction biomass, the degradation of the hemicellulose and cellulose components are larger through the pre-treatment process. As a result, torrefaction biomass has a greater grindability compared with ashless biomass.” è We appreciate your valuable comments for our manuscript. We agree with the reviewer’s opinion. In the discussion chapter 3.4, we added the paragraph explaining the grindability for pre-treatment methods in this study while deleting the previous sentence as below.

There is a mistake in figure 5. T.PSD present biggest particles than R.PSD. it is not the case in the cumulative distribution (figure 6). Figure 5 is very difficult to understand, I think that the figure 6 is sufficient to support the discussion.(a) è We appreciate your valuable comments about our statements. As per your comments, we divided Figure 5 obtained during the experiment into two (a, b) and added data.

(b)

Reviewer 2 Report

The paper submitted presents the results of comparing two pretreatments over a Woody and an herbaceous biomass. In particular, both pretreatments are torrefaction and deashing with an acetic acid solution. The paper is well written but the significance and novelty of the results are low. The results are quite expected as there are already works about torrefaction and acid-washing of biomass. The authors should emphatize more strongly the relevance of the paper to be reconsidered for publication. Other comments are:

Title of the manuscript: it is not sufficiently clear about the objective of the paper. The authors should clarify what is the purpose of such pretreatments, I mean, the final application of pretreated biomass. Section 3.1: "Therefore, the proximate constituents of torrefied samples became closer to coal samples. In contrast, A. PSD and A. Kenaf produced by the ashless process showed a slight increase in VM and a slight decrease in FC compared to R. PSD and R. Kenaf". This is a too obvious result as ashless process removes inorganics (non volatile) from biomass and, consequently, the Volatile Matter will increase in proportion. Figure 2: to represent pore size distributions, the authors should present the derivative curves and indicate which methof they have Applied. It is very probable that the method Applied (not specified) is not suitable for microporosity determination. Figure 5: very confusing. Too many bars.

Author Response

Review on the article

“Comparing the Physicochemical Properties of Upgraded Biomass Fuel by Torrefaction and the Ashless Technique”

General comment _2

The paper is well written but the significance and novelty of the results are low. The results are quite expected as there are already works about torrefaction and acid-washing of biomass. The authors should emphatize more strongly the relevance of the paper to be reconsidered for publication.è Following sentences have added on page 2 in revised manuscript. “Therefore, the ashless biomass technique used in this study is a state of the are pre-treatment method to remove ash and alkali matter in herbaceous as well as woody biomass. In addition, it is commercially applicable for the production of ashless biomass, which is seeing increasing potential as an acidic treatment of biomass compared to other pretreatment methods. To the best of our knowledge, detailed studies on the physicochemical properties of ashless biomass produced by the above method have not been performed and compared with other pretreatment methods thus far.”   è We appreciate your valuable comments for our manuscript. As per your comments, we emphasize more relevance of this paper in the introduction section as follows; Title of the manuscript: it is not sufficiently clear about the objective of the paper. The authors should clarify what is the purpose of such pretreatments, I mean, the final application of pretreated biomass.  è We agree with the reviewer’s opinion. We changed the paper title and chapter title of the manuscript. Section 3.1: "Therefore, the proximate constituents of torrefied samples became closer to coal samples. In contrast, A. PSD and A. Kenaf produced by the ashless process showed a slight increase in VM and a slight decrease in FC compared to R. PSD and R. Kenaf". è We appreciate your valuable comments for our manuscript. In this sentence, we tried to show different trends in VM and FC after ashless process, unlike torrefaction. As per your comment, we have modified this sentences as follows;  è Following sentences have added on page 5 in revised manuscript. “In contrast to torrefaction, A. PSD and A. Kenaf produced by the ashless process showed increase in VM and decrease in FC compared to R. PSD and R. Kenaf because of reduced ash contents and impact of ash rejection process.” This is a too obvious result as ashless process removes inorganics (non volatile) from biomass and, consequently, the Volatile Matter will increase in proportion. Figure 2: to represent pore size distributions, the authors should present the derivative curves and indicate which methof they have Applied. It is very probable that the method Applied (not specified) is not suitable for microporosity determination. The specifications for ASAP 2020 plus physisorption (Micromeritics) are as follows: pressure measurement range: 0 to 950 mmHg, resolution: up to 0.1 mmHg transducer, and accuracy: ± 0.1%. This system includes a 0.1 mmHg transducer and a high vacuum pump. This system can provide porosity data for micropores with a size of 0.35 to 3 nm as well as for pores of a larger size. Therefore, this instrument was used in the investigation of nanomaterials in this study [18-20].”.[19] Buyi Li, Xin Huang, Liyun Liang and Bien Tan, Synthesis of uniform microporous polymer nanoparticles and their applications for hydrogen storage, J. Mater. Chem., 2010, 20, 7444–7450  (b)   (a) è We appreciate your valuable comments for our manuscript. As per your comments, the figure has been changed clearly as follows; [20] Arash Arami-Niy, Wan Mohd Ashri Wan Daud, Farouq S. Mjalli, Comparative study of the textural characteristics of oil palm shell activated carbon produced by chemical and physical activation for methane adsorption, Chemical Engineering Research and Design, 2011, 89(6) 657-664 [18] Arash Arami-Niya, Wan Mohd Ashri Wan Daud, Farouq S. Mjalli, Using granular activated carbon prepared from oil palm shell by ZnCl2 and physical activation for methane adsorption, Journal of Analytical and Applied Pyrolysis, 2010, 89(2) 197-203 è We appreciate your valuable comments for our manuscript. The specifications for ASAP 2020 plus physisorption (Micromeritics) are as follows; Figure 5: very confusing. Too many bars.  è We appreciate your valuable comments about our statements. As per your comments, we divided Figure 5 obtained during the experiment into two and added data.

(a)

(b)

Round 2

Reviewer 1 Report

Thank you for your answer and for the added informations.

Author Response

Featured Application: Torrefaction and ashless techniques in this work are used to upgrade physicochemical properties of woody and herbaceous biomass. The ashless technique combining torrefaction process is providing promising application in herbaceous biomass specially.

Reviewer 2 Report

The authors have taken into consideration the reviewer's comments and improved the manuscript. It can be accepted for publication.

Author Response

(The authors gave the same response as above.)
